# Hydroxypropyl Methylcellulose Bioadhesive Hydrogels for Topical Application and Sustained Drug Release: The Effect of Polyvinylpyrrolidone on the Physicomechanical Properties of Hydrogel

**DOI:** 10.3390/pharmaceutics15092360

**Published:** 2023-09-21

**Authors:** Patrick Pan, Darren Svirskis, Geoffrey I. N. Waterhouse, Zimei Wu

**Affiliations:** 1School of Pharmacy, Faculty of Medical and Health Sciences, The University of Auckland, Auckland 1142, New Zealand; t.pan@auckland.ac.nz (P.P.); d.svirskis@auckland.ac.nz (D.S.); 2School of Chemical Sciences, Faculty of Science, The University of Auckland, Auckland 1142, New Zealand; g.waterhouse@auckland.ac.nz

**Keywords:** hydrogels, factorial design, controlled release, topical delivery, texture analysis, rheology, bioadhesion, intermolecular interactions, hydroxypropyl methylcellulose, polyvinylpyrrolidone

## Abstract

Hydrogels are homogeneous three-dimensional polymeric networks capable of holding large amounts of water and are widely used in topical formulations. Herein, the physicomechanical, rheological, bioadhesive, and drug-release properties of hydrogels containing hydroxypropyl methylcellulose (HPMC) and polyvinylpyrrolidone (PVP) were examined, and the intermolecular interactions between the polymers were explored. A three-level factorial design was used to form HPMC–PVP binary hydrogels. The physicomechanical properties of the binary hydrogels alongside the homopolymeric HPMC hydrogels were characterized using a texture analyzer. Rheological properties of the gels were studied using a cone and plate rheometer. The bioadhesiveness of selected binary hydrogels was tested on porcine skin. Hydrophilic benzophenone-4 was loaded into both homopolymeric and binary gels, and drug-release profiles were investigated over 24 h at 33 °C. Fourier transform infrared spectroscopy (FTIR) was used to understand the inter-molecular drug–gel interactions. Factorial design analysis supported the dominant role of the HPMC in determining the gel properties, rather than the PVP, with the effect of both polymer concentrations being non-linear. The addition of PVP to the HPMC gels improved adhesiveness without significantly affecting other properties such as hardness, shear-thinning feature, and viscosity, thereby improving bioadhesiveness for sustained skin retention without negatively impacting cosmetic acceptability or ease of use. The release of benzophenone-4 in the HPMC hydrogels followed zero-order kinetics, with benzophenone-4 release being significantly retarded by the presence of PVP, likely due to intermolecular interactions between the drug and the PVP polymer, as confirmed by the FTIR. The HPMC–PVP binary hydrogels demonstrate strong bioadhesiveness resulting from the addition of PVP with desirable shear-thinning properties that allow the formulation to have extended skin-retention times. The developed HPMC–PVP binary hydrogel is a promising sustained-release platform for topical drug delivery.

## 1. Introduction

Topical drug delivery attracts significant attention due to its potential to provide targeted and localized therapy for various dermatological conditions. However, the efficacy of topical formulations can often be compromised by factors such as poor retention on the skin and low drug penetration into the skin, leading to the need for frequent application [1]. The outermost layer of the skin, the stratum corneum, constitutes a strong barrier and makes it difficult for drug molecules to penetrate and cross the skin at clinically relevant rates. An increase in the contact time of topically applied formulation allows for a higher quantity of the active agents to eventually be delivered. In dermatological conditions such as in psoriasis, a controlled-release formulation will allow for the local release of a drug over a prolonged period, reducing the frequency of administration, and improving patient compliance and clinical outcomes. Controlled-release technologies using bioadhesive gels provide a solution to overcome these challenges and improve the effectiveness of topical drug delivery [2,3,4,5]. These gel formulations can regulate the release rate of drugs over time. They can be designed to provide sustained drug exposure to the target site, reducing the risk of systemic side effects, and provide flexibility in dosing regimens to match the specific requirements of different dermatological conditions [2,3,4,5]. Thus, topical controlled-release gel formulations offer significant advantages and hold great promise in advancing the field of dermatological drug delivery.

Among the various topical controlled-release formulations, hydrogels have been extensively explored as an effective medium for sustained topical drug delivery [2,3,4,5]. Hydrogels are homogeneous semisolids which consist of a water-swollen hydrophilic polymer three-dimensional network possessing a high water content [6]. Hydrogels offer numerous benefits as a controlled-release medium, including the ability to quickly dry and form a thin-film that is non-greasy and non-occlusive, whilst also being cosmetically elegant and easy to apply [2,7]. Such systems have been used to locally deliver anti-inflammatories and anesthetics such as borneol, curcumin, and lignocaine during the treatment of skin wounds [3,5,8], where an extended local retention time is desirable. Furthermore, polymers in gels ensure good film formation and stability on the skin, as well as good water and sweat resistance [9].

Various physicomechanical properties (rheological and mechanical properties) of a gel-based topical formulation determine its retention, penetration, and drug-release rates. Viscosity of the gel medium plays an important role in increasing the retention time of the formulation on the skin while prolonging drug release [3,4]. Bioadhesiveness is another important factor to consider when designing topical controlled-delivery formulations where extended skin contact is required for the drug to be delivered to the target site over a period of time [10,11]. The main mechanism of bioadhesion is intermolecular bonding, and, for many materials, it is due to the formation of interfacial hydrogen bonds between the adhesive gel and biological surface [12,13].

With hydrogels, the low polymer and high water content allows for soft, deformable, and flexible networks that can accommodate skin movement [14,15], making them desirable for topical drug delivery. However, the adhesiveness of hydrogels is generally low due to the majority of their volume being composed of polar water molecules that do not actively participate in joining materials [15]. To overcome this, combinations of different hydrogel polymers have been explored to boost hydrogel adhesion [14,16,17], resulting in “binary double network-like gels” or “binary gels”. However, the addition of the secondary polymer often causes an increase in gel viscosity or decreases the spreadability of the gel on the skin, necessitating the lowering of polymers concentrations, which in turn can reduce adhesiveness [18].

Hydroxypropyl methylcellulose (HPMC) is a commonly used hydrophilic polymer in controlled-release formulations due to its thickening, gelling, and swelling properties, which can form highly stable, clear, and odorless hydrogels [19]. Cosmetically, HPMC gels provide a thick but non-tacky feel, produce a strong and flexible film upon drying, disperse easily on the skin, have a cooling effect, and are non-comedogenic [20]. The bioadhesive property of HPMC has been attributed to the presence of abundant -OH functional groups in the molecule that can form hydrogen bonds with water and other HPMC molecules [21] (Figure 1A). Additionally, HPMC has a minimal interaction with drugs (other than H-bonding interactions) and has demonstrated the ability to improve bioadhesion and enhance local delivery of drugs through improved retention [22,23]. Polyvinylpyrrolidone (PVP), a hydrophilic synthetic polymer, is commonly used in controlled-drug-delivery systems due to its biocompatibility [24,25]. However, the direct use of pure PVP hydrogels is limited due to their low swelling capacity and poor mechanical properties. Therefore, PVP is often blended with different polymers to improve the physicomechanical properties of the preparations according to the requirements of the application [24,26]. Of note, PVP polymers have excellent adhesive properties due to the abundance of carbonyl groups (Figure 1B) that can establish hydrogen bonds with biological surfaces, making PVP an ideal component in bioadhesive delivery systems [25,27,28,29]. Furthermore, binary hydrogels can form a strong cross-linked film that can adhere to the skin while maintaining a smooth feel, making them ideal for topical formulations for controlled drug delivery [27,30]. Little information is presently available for HPMC–PVP binary hydrogels; however, the addition of PVP has previously been reported to reduce the tackiness of HPMC solutions, as a result of a net reduction in the hydrogen-bonding network between the HPMC chains which is caused by PVP addition [31]. We hypothesized that the HPMC–PVP system may form hydrogels with a high adhesiveness without significantly impacting other mechanical parameters such as viscosity and spreadability, offering a platform for sustained topical drug delivery.

This research aimed to evaluate the effectiveness of using binary HPMC and PVP hydrogels for sustained topical drug delivery. The mechanical properties of the gels were characterized using a texture analyzer and a Brookfield rheometer, and bioadhesiveness was further tested on porcine skin. With the aid of a three-level two-factor (3^2^) factorial design, the effects of the addition of PVP on the mechanical parameters of the HPMC-PVP hydrogel’s viscosity, bioadhesiveness, and drug-release rates were investigated to determine if an optimal binary gel formulation can produce a topical formulation that is more suited to the sustained release of a drug (benzophenone-4 in this case) into the skin.

## 2. Materials and Methods

Hydroxypropyl methylcellulose K100 (HPMC), polyvinylpyrrolidone K25 (PVP), benzophenone-4, sodium phosphate dibasic, sodium phosphate monobasic, and dialysis bags from regenerated cellulose membranes with a molecular weight cut-off (MWCO) of 14,000 Da were all purchased from Sigma-Aldrich (Auckland, New Zealand). The HPMC and PVP powders were dried overnight at 60 °C prior to use and were stored in a desiccator. Sodium phosphate dibasic and sodium phosphate monobasic were used to prepare 0.1 M of phosphate buffered saline (PBS), and the final pH was adjusted to 5.5 using hydrochloric acid. Milli-Q water was obtained from a Millipak^®^ 0.22 μm system (Millipore Corporation, Bedford, MA, USA). All other reagents were of analytical grade and purchased from Sigma-Aldrich (Auckland, New Zealand).

Fresh porcine skin (aged 5–6 months) from the flank was obtained from a local abattoir (Auckland Meat Processes, Auckland, New Zealand).

### 2.1. Preparation of Polymeric Hydrogel Systems

#### 2.1.1. HPMC and PVP Homopolymeric Hydrogels

Homopolymeric gel formulations were prepared using HPMC alone at eight different concentrations (2%, 4%, 6%, 8%, 10%, 12%, 14%, and 16%, *w*/*w*) or PVP alone at three different concentrations (3%, 6%, and 9%, *w*/*w*). Hydrogels were prepared following a dry-blending method [32]. Briefly, the required amounts of dry HPMC or PVP were weighed and dispersed in half the necessary amounts of PBS (0.1 M, pH 5.5, pre-warmed to 80 °C), followed by vigorous stirring for 10 min to obtain a well-dispersed mixture. Further PBS (at room temperature) was then slowly added to produce the final desired concentration. The mixture was stirred for another 10 min at room temperature, followed by cooling for 10 min in an ice bath. The gels were then sealed and stored at 4 °C for at least 48 h to ensure the complete hydration of the polymers and to allow the escape of the entrapped air bubbles.

#### 2.1.2. HPMC–PVP Binary Hydrogels and Factorial Design Analysis

The dry-blending method was used to prepare the binary gel formulations. Briefly, appropriate amounts of dry HPMC and PVP powders were uniformly mixed before the addition of PBS and mixing well to obtain uniform hydrogels.

A three-level two-factor (3^2^) factorial design was used to generate the binary gel formulation (Table 1) and to assess the effect of the polymer concentrations in the HPMC–PVP binary hydrogels on their physicomechanical properties. Based on the properties of the homopolymeric HPMC gels, the concentrations of HPMC (X_1_, 4, 8 and 12% *w*/*v*) and PVP (X_2_; 3, 6, 9% *w*/*v*) were considered as independent variables. The response variables Y including adhesiveness and viscosity (at shear rate of 4 s^−1^) were of interest.

### 2.2. Characterization of Hydrogels

#### 2.2.1. Texture Profile Analysis

The mechanical properties of the homopolymeric and binary hydrogels were investigated using the TA.XT Plus texture analyzer (Stable Micro System, Surrey, UK). A 2 kg loading cell and a cylindrical stainless-steel probe (diameter 25 mm) were used for all measurements. The gel samples (50 g) were placed into glass jars to produce a cylindrical gel mass (diameter 50 mm × 80 mm height) and stored at room temperature for 12 h prior to testing.

During the texture profile analysis, the probe was compressed twice into each gel sample at a defined rate of 1 mm·s^−1^ with a trigger force of 0.001 g, during which the probe would penetrate to a depth of 10 mm into the gel sample. There was a delay period of 15 s between the end of the first and the beginning of the second compression. At least three replicates were performed for each formulation at ambient temperature (21 ± 2 °C) using fresh samples in each case. The data collection and analysis were performed using the Texture Exponent 3.0.5.0 software package provided with the instrument (Stable Micro System, Surrey, UK). The force–time graphs (typically as shown on Figure 2B,C) were recorded for the determination of the mechanical parameters, namely hardness, compressibility, adhesiveness, and cohesiveness [33].

#### 2.2.2. Rheological Characterization

The rheological properties of the gel formulations containing different concentrations of polymers were analyzed using a rotational Brookfield DV-III+ cone and plate rheometer (Brookfield Engineering Laboratories Inc., Middleborough, MA, USA). The rheometer was fitted with a Flat Plate SST ST 40 mm diameter spindle and was operated by the Brookfield Rheocalc operating software version 3.2.47. The sample temperature was controlled at 33 ± 0.1 °C.

For each measurement, 50 μL of gel was carefully pipetted using a large-ended pipette tip, with the tips cut off, to ensure uniform sampling of gel. The sample was applied to the lower chamber of the viscometer avoiding any air bubbles and was allowed to equilibrate for at least 5 min before analysis. The samples were subjected to continuous shear analysis and a logarithmic sweep was performed at shear rates of 4–300 s^−1^. Each speed was maintained for 30 s to allow for data collection. Three replicates were performed for each formulation, and viscosity curves at shear rates were plotted to understand the flow properties.

### 2.3. Ex Vivo Bioadhesion Testing

Porcine skin from the flank was used to assess bioadhesiveness of the selected binary hydrogel which contained HPMC 12% and PVP 6% (denoted as H12P6).

#### 2.3.1. Tissue Preparation

The elapsed time from the slaughter of the pig to the removal of the skin was approximately 2 h. The skin sections were stored in normal saline (0.9%) to prevent dehydration during transport to the laboratory for dissection. The damaged or bruised sections were discarded. A scalpel was used to remove the subcutaneous tissue and the remaining full layer of the stratum corneum with epidermis was then cut into either a circular section (diameter of 25 mm) for use as the attached skin, or square sections (30 mm × 30 mm) to be used as the substrate in bioadhesion testing. These length and width measurements reflect the skin in its relaxed state, where the wrinkles and folds were not stretched or flattened. The prepared skin sections were immediately frozen in liquid nitrogen and stored at −20 °C for no longer than 4 weeks before use.

Porcine skin from the flank, as opposed to the standard porcine ear [34], was used to assess bioadhesion. Evidence has supported that it is an effective in vitro substrate for simulating human skin in terms of histological and physiological properties [35], while also providing the benefit of being able to test using larger sections of skin.

#### 2.3.2. Bioadhesion Analysis

The excised skin sections were thawed overnight at 4 °C and soaked in phosphate-buffered saline (PBS) (0.1 M, pH 5.5) for 10 min. The excess surface moisture was removed by blotting with filter paper after placing a 2 kg weight over the skin for 5 min. Each piece of skin was used only once for each gel preparation.

The bioadhesive force between porcine skin and binary gel was assessed using the TA.XT Plus texture analyzer in Hold-Until-Time mode. To identify differences in bioadhesion results, two setups were used: one testing the steel probe with a single skin substrate, and a second setup testing between two skin sections (Figure 2A).

A total of 1 g of gel was spread homogenously over the entire surface of the substrate skin sections (30 mm × 30 mm), which were secured onto a petri dish with double-sided tape (3M Scotch Mount^TM^). The circular skin sections were attached using double-sided tape (3M Scotch Mount^TM^) to the lower end of the cylindrical probe (diameter 25 mm), facing downward and opposing the substrate skin sections.

The upper part of the texture analyzer (with the attached skin) was placed as close as possible to the substrate skin. Contact was avoided between the two skin sheets. In this position, the texture analyzer was lowered to 0.1 mm·s^−1^, which has been shown to give the best discriminative values [36], until contact between the substrate skin and the attached skin was made. The triggering force (by which the contact with the sample was calculated) was 0.01 N. The two skin pieces were in contact for 15 s under a force of 0.5 N. The upper part of the texture analyzer was lifted at a speed of 0.1 mm·s^−1^ until the separation of the two skin sheets occurred.

The tests were conducted at ambient temperature (21 ± 2 °C) and each experiment was replicated at least three times using a fresh sample of gel. The mucosa was gently cleaned with saline-soaked damp tissue before testing each replicate.

Peak tension can be derived from a force–time graph as the maximum force required to separate the adhesive interface. The area under the force–distance curve during the separation of the hydrogels from the skin surface is regarded as the work of adhesion (Figure 2C).

### 2.4. Drug Release

The dialysis tubing method [37] was used to investigate the drug release from various gel systems.

Drug-loaded gels were prepared during preparation of the base gel as described above, with the drug pre-added to the PBS phase. The final concentration of benzophenone-4 in each gel was 25 mg/g (2.5%). Drug-loaded gels (2 g) were then packed into 3 mL syringes and loaded into prepared dialysis tubing before sealing with dialysis clips. Care was taken to pack gels tightly, with no air bubbles within the tubing, to form a gel column (diameter 20 mm × 25 mm height), providing a total surface area for diffusion of ~3140 mm^2^.

Four formulations (PVP 6%, HPMC 12%, HPMC 13%, H12P6) suspended in PBS and one formulation (HPMC 13% in MeOH) in methanol were tested for drug release by placing the dialysis bags loaded with samples in 50 mL of either PBS or methanol as external media in separate Falcon tubes and suspended in a water bath at 33 °C with oscillation set at 60 rpm. Aliquots (1 mL) from the external media were withdrawn at various time points (0.25, 0.5, 0.75, 1, 2, 3, 4, 5, 6, and 24 h) and immediately replaced with an equivalent volume of PBS or methanol. The drug content in the external media was analyzed directly through a validated high-performance liquid chromatography (HPLC) assay [38].

### 2.5. Fourier Transform Infrared Spectroscopy

To reveal and identify any intermolecular interactions between functional groups, infrared transmission spectra of pure benzophenone-4 and freeze-dried hydrated gel samples of HPMC, PVP, and HPMC–PVP binary gel mixtures (1:1 weight ratio) with and without benzophenone-4 were obtained using a Fourier transform infrared (FTIR) spectrophotometer (Bruker Alpha Eco-ATR FTIR Spectrometer; OPUS 8.7.41 ALPHA, Mannheim, Germany). The samples were analyzed above a diamond crystal using the ATR mode. The spectra were collected over the wavenumber range 4000 to 400 cm^−1^ at resolution of 4 cm^−1^ to investigate possible interactions between the active functional groups of benzophenone-4 with PVP and HPMC.

The optical spectroscopy software, Spectragryph, Version 1.2.16.1, was used to visualize and manipulate the ATR-FTIR spectra [39]. A baseline correction was applied to all spectra using the standard normal variate approach.

### 2.6. Stability Studies

The prepared gels (HPMC gels and HPMC–PVP binary gels) were packed into Eppendorf tubes (2 mL) and stability studies were performed in the temperature and relative humidity (RH) conditions stipulated by the International Council for Harmonisation of Technical Requirements for Pharmaceuticals for Human Use. The samples were stored at 4 °C, 25 °C (60% RH), 30 °C (65% RH), and 40 °C (75% RH) for a period of 3 months. The samples were withdrawn at 15-day intervals and evaluated for physical appearance, pH, and viscosity. The drug content was evaluated at 3 months [40].

### 2.7. Statistical Analysis

All data are expressed as the mean ± standard deviation (SD). A statistical analysis of each parameter of interest was carried out using Student’s *t*-test for dependent and independent samples, a one-way analysis of variance (ANOVA) and Tukey’s HSD post hoc test. A *p*-value < 0.05 was considered significant. All statistical calculations were performed using the GraphPad Prism for Windows (Version 9, GraphPad Software, La Jolla, CA, USA).

## 3. Results

### 3.1. HPMC Homopolymeric Hydrogels

#### 3.1.1. Texture Profile

The hardness, compressibility, and adhesiveness force all showed an exponential relationship to the concentration of HPMC in the homopolymeric HPMC gels (Figure 3A–C). The cohesiveness was largely unaffected by the HPMC concentration (2–16%) (Figure 3D), indicating that the intermolecular forces between the HPMC polymer chains were independent of polymer concentration.

The homopolymeric PVP presented as a free-flowing solution with concentrations of 3%, 6%, and 9%, *w*/*w*, thus no texture profile analysis could be performed. This was expected, owing to the low swelling capacity of the PVP and thus its inability to establish a viscous hydrogel network [26].

#### 3.1.2. Rheological Properties

The rheological data showed that increasing concentrations of HPMC led to an exponential increase in viscosity, as measured at low shear rate (Figure 4B). The HPMC gels from 2% to 8% exhibited near-Newtonian flow behavior, as evidenced by the almost-linear trend in the rheograms (Figure 4B,C). When the HPMC polymer concentrations increased above 8%, the HPMC hydrogels displayed non-Newtonian (shear-thinning) behavior.

### 3.2. HPMC–PVP Binary Hydrogel and Factorial Design Analysis

#### 3.2.1. Texture Profile

The adhesiveness of the resulting binary gels increased as a function of the concentration of both polymers. The addition of PVP at 0, 3, 6, or 9% (*w*/*w*) to the HPMC gels (4–12%, *w*/*w*) increased adhesiveness (Figure 5C) while having a minimal impact on hardness and compressibility (Figure 5A,B). For example, the homopolymeric gel HPMC 12% (H12P0) had an adhesiveness of 40.96 ± 0.37 g·s. The addition of PVP 6% (H12P6) to create a binary gel increased the adhesiveness to 253.64 ± 14.87 g·s, representing a 6.19-fold increase (Figure 5E). At higher concentrations of PVP 9% (H12P9), a higher adhesiveness of 405.08 ± 13.17 g·s was achieved. However, hardness and compressibility, undesirable properties for a topical formulation, also increased significantly for H12P9.

The binary H12P9 formulation exhibited a sharp decrease in the cohesiveness of the gel (Figure 5E), indicating a compromised structure of the polymer network. Thus, this binary gel formulation was excluded from further studies. The results suggested that the H12P6 gel formulation was the most appropriate for further studies examining drug release due to its high adhesiveness, along with acceptable hardness, compressibility, and cohesiveness.

#### 3.2.2. Rheological Properties

Stronger shear-thinning properties were observed with increasing polymer concentrations in the hydrogels (Figure 6A). Increasing the concentration of PVP in the HPMC–PVP binary hydrogels lead to an increase in viscosity, together with more profound shear-thinning properties. The viscosity for all gels dropped to similar levels when the shear rate was increased to 200 s^−1^ (Figure 6B). Interestingly, despite the increase of PVP to 9% for H12P9, there was a slight decrease in the viscosity along with a significant reduction in its’ peak shear stress (at shear rate = 86 s^−1^), which was lower than both H12P3’s and H12P6’s (Figure 6A). This is aligned with a reduction in the cohesiveness for the same formulation which was seen in the texture analyzer tests, and it further reinforces the finding that the structure of the polymer network in the binary hydrogels was compromised at high PVP concentrations.

At modest shear rates of 300 s^−1^, all formulations achieved a very low viscosity value, which makes them suitable for easy application onto the skin.

#### 3.2.3. Factorial Design Analysis

The two-dimensional contour, surface, and interaction plots for adhesiveness and viscosity are shown in (Figure 7). The plots clearly demonstrated that neither adhesiveness nor viscosity increased proportionally to the HPMC concentration in the binary gels. The results also revealed that the increase in adhesiveness and viscosity caused by increasing the PVP concentration from 3% to 6% was only apparent at high concentrations of HPMC (12%). However, further increasing the PVP to 9% increased adhesiveness but not the viscosity.

### 3.3. Ex Vivo Bioadhesiveness

Figure 8 shows bioadhesion results for the binary gel formulation H12P6. This formulation showed the highest adhesiveness, yet an appropriate viscosity for topical application. Significant differences were observed in the peak force and work of adhesion between the two setups used (either single or double skin substrates), but not adhesiveness.

A significantly large drop in the peak force for both the homopolymeric and binary gels was observed between the single surface skin setup and the double skin setup, possibly attributable to the larger volume of skin that was displaced to reach the specified contact force of 0.5 N. This was also accompanied by a small but non-significant decrease in the adhesiveness and work of adhesion in the double substrate skin setup.

Compared with homopolymeric 12% HPMC gel, the hardness remained the same in the binary gel with addition of 6% PVP. However, a significant increase in the adhesiveness and work of adhesion were observed in the binary gel (*p* < 0.01, *p* < 0.05) using method one. No differences between the two gel formulations were observed using method two. 

### 3.4. Drug-Release Profiles

The cumulative drug release of benzophenone-4 from different gel formulations, including a free drug solution as a control, is illustrated in Figure 9. Release from the drug solution was complete within 3 h, indicating that the dialysis tubing did not impact drug release and that sink conditions were present throughout. Even though it possessed a low viscosity, the homopolymeric PVP 6% impeded the diffusion of benzophenone-4 out of the gel matrix into the external media. The cumulative release amount of benzophenone-4 (Q_t_) was linearly dependent on the square root of the time (t), implying that the release kinetics followed a Higuchi model (R^2^ = 0.959). In contrast, all HPMC gels demonstrated zero-order release kinetics in the first 6 h (R^2^ > 0.983 to 0.989). The release rate from the 13% HPMC gel was 7.2% (of total dose) h^−1^, which increased to 8.9% h^−1^ as the HPMC polymer concentration was reduced to 12%, with the increased release being attributed to the reduced viscosity (52.8 Pa·s for 12% HPMC gel vs. 76.4 Pa·s of 13% HPMC gel; at shear rate of 4 s^−1^, as shown in Figure 4C). No significant difference in drug release was observed between the PBS and methanol (MeOH) as the external media, indicating that the differences in solubility of benzophenone-4 in the external media did not impact drug release and that sink conditions were present throughout. For all HPMC gels, the drug-release rate after 6 h became slower. Of note, no benzophenone-4 was detected in the external media for the binary H12P6 formulation over 24 h, indicating that the rate of drug release was very low and that the total amount of drug released below the limit of detection (LOD) of 1.08 μg/mL (0.11% of total dose).

### 3.5. Fourier Transform Infrared Spectroscopy

The FTIR spectra for the various samples are shown in Figure 10. The benzophenone-4 (BNZ4) showed characteristic peaks around 1061 cm^−1^ and 599 cm^−1^ due to S=O and S-O stretching modes, respectively, in the sulfonic acid (-SO_3_H) group of BNZ4, as well as further peaks in the 1600–1000 cm^−1^ region due to C=O, aromatic C=C, and C-O stretching. The FTIR spectrum of HPMC was dominated by an intense feature at 1049 cm^−1^ due to C-O stretching, along with further peaks at 3395 cm^−1^ and 1373 cm^−1^ due to O-H stretching and C-O-H bending vibrations of hydroxyl groups, respectively. Similar peaks were reported by other groups for HPMC [41]. The PVP showed an intense absorption band at 1654 cm^−1^, which could readily be assigned to an amide C=O stretching mode [42]. The spectrum for the freeze-dried gel comprising HPMC–PVP (1:1 weight ratio) showed no shift in the characteristic peaks of each polymer, suggesting that no strong intermolecular interactions occur between the two polymers in the dry state. In the PVP + BNZ4 and HPMC–PVP + BNZ4 spectra, the S=O stretching modes for BNZ4 were observed at 1074 cm^−1^ (cf. 1061 cm^−1^ for pristine BNZ4). The same shift was not seen in HPMC + BNZ4, as this peak was obscured by the intense C-O stretching mode of HPMC which occurs at a similar frequency to the S=O stretch of BNZ4. In the low frequency region, HPMC + BNZ4, PVP + BNZ4, and HPMC–PVP + BNZ4 all showed a sharp peak around 605-607 cm^−1^ (assigned to a S-O stretching mode of the -SO_3_H or -SO_3_^−^ groups), which again was at higher frequencies compared to the same mode for BNZ4 (599 cm^−1^). Results imply that the BNZ4 interacted with the PVP and through the sulfonate functional group.

### 3.6. Stability Studies

All gel and binary gel formulations were found to be stable over a 3-month period. No significant changes in physical appearance, pH, viscosity, or drug content were observed.

## 4. Discussion

Hydrogels are an increasingly popular choice in topical applications, particularly in the controlled delivery of drugs through the skin. However, their low adhesiveness can limit their efficacy, and attempts to increase adhesion can negatively impact other properties such as hardness, compressibility, and viscosity. This study aimed to investigate the use of HPMC and PVP binary hydrogels to improve the adhesion of gel-based drug delivery formulations without compromising other texture properties. The results of the study demonstrated that binary hydrogels prepared using HPMC and PVP can improve adhesiveness without significantly impacting other mechanical properties. Other studies that also utilized a binary mixture of polymers, such as HPMC and polycarbophil, have demonstrated similar results in improving bioadhesive properties [43].

The ability to improve adhesiveness without significantly affecting other properties allows for the formulation of highly adhesive topical products to increase the retention time of the drug on the skin for controlled drug delivery, while not negatively affecting the cosmetic acceptability of the gel. A binary formulation (H12P6) was selected for studies on drug release and ex vivo bioadhesiveness, as it exhibited the highest adhesiveness without compromising other mechanical properties. HPMC 12% was used as a reference. Further increasing the PVP concentration from 6% to 9% caused a large reduction in the cohesiveness of the binary gel, indicating a compromised structural integrity of the polymer matrix. This is similar to findings in the literature, where it is suggested that PVP may interact with HPMC chains in the aqueous medium and consequently reduce the extent of HPMC–HPMC bonding [31]. Our results show that no change in gel cohesiveness occurred up to 6% PVP, suggesting that significant intermolecular bond interference only occurs above a critical concentration threshold where there are sufficient PVP molecules to obstruct HPMC–HPMC bonding interactions.

In addition to their texture properties, the rheological properties of binary hydrogels were also investigated. The study found that binary hydrogels exhibit pseudoplastic behavior (shear-thinning), which is desirable for topical products as it ensures uniform distribution on the skin and ease of application or good spreadability [43]. At modest amounts of shear stress (300 s^−1^), the viscosity of all formulations was extremely low (<10 Pa·s), suggesting that the formulations would be easy to apply in practical settings. The shear stress on application of topicals was in excess of 10^4^ s^−1^ [44].

This study also identified the ability of the H12P6 binary gel to “lock” in hydrophilic drugs, with benzophenone-4 remaining below the LOD of the analytical method after 24 h in drug-release studies. This demonstrates that the drug cannot access the matrix surface through the wetted pore network into the external media (PBS or methanol). In contrast, the HMPC homopolymeric hydrogels provided a sustained release with a nearly zero-order kinetic profile. Drug-release kinetics from hydrophilic matrices depend on several processes including the swelling of the polymer, water penetration through the matrix, dissolution of the drug, transport of the drug through the swollen matrix, and erosion of the matrix itself [45,46]. The most commonly used mathematical models to study cumulative drug-release behavior in hydrogels include zero- and first-order kinetic models, Higuchi, Peppas, and Hixon–Crowell models [46]. However, none are suitable for modelling the low drug release from the H12P6 binary gel, as it does not seem to be significantly influenced by the physical properties, such as viscosity, of the gel. This indicates that the binary gel lacks a highly structured matrix due to the low degree of polymer swelling and factors such as drug diffusion, mesh size, and erosion would then not be limiting factors to drug release [46]. Percolation theory is also unsuitable for explaining the drug release of the binary gel owing to the extremely low porosity and lack of connectivity and permeability [45,47,48], indicating that more complex processes such as drug–matrix interactions are influencing the release of the drug. Alternatively, one more explanation for the slow release could be due to the shear-thinning properties that made the gels in the dialysis bags extremely viscous at static conditions, thus impeding drug diffusion and erosion of the gel.

It was reported that the addition of PIP impairs the intermolecular binding of the HPMC polymer chains, thus leading to a reduction in viscosity [31]. Indeed, as shown in Figure 4A and Figure 6B, the addition of PVP to the HPMC gels slightly increased the viscosity, particularly when shear rate was low. The 3^2^ factorial design (Figure 7) also confirms the relatively smaller impact of the PVP on these properties compared with that of the HPMC. A similar complex is seen in polyvinylpyrrolidone-iodine [49]. The PVP polymer arranges itself such that a proton is fixed via a short hydrogen bond between two carbonyl groups of two adjacent pyrrolidone rings, forming a positive charge where a negatively charged molecule, such as a triiodide anion, can be bound ionically (Figure 11A) [49]. Benzopohenone-4 may interact with this positively charged area and become retained in the polymer matrix rather than releasing freely to the external media. In the PBS buffer used in this study, the sulfonic acid group (-SO_3_H) of benzopohenone-4 would be deprotonated and exist as the sulfonate ion (-SO_3_^−^). The blue-shift in the S=O and O-S-O peaks of benzopohenone-4 after interactions with the polymers may have been the result of deprotonation of the sulfonic acid group. This would create the possibility of electrostatic interaction between benzophenone-4 and a hydrogen atom between two carbonyl groups of the PVP polymer (Figure 11B). Such electrostatic interactions might account for the very strong binding of benzopohenone-4 in the H12P6 binary gel that led to the slow release.

Finally, this study highlights the lack of a standardized method for measuring the work of adhesion of hydrogels to skin substrates. The results showed that the peak force, adhesiveness, and work of adhesion values used to assess bioadhesion differed substantially depending on whether one or two skin substrates were used. When testing bioadhesion parameters using a double skin substrate setup, a large reduction in the peak force was seen, alongside non-significant decreases in the adhesiveness and work of adhesion. This contrasts with the expected increase due to the larger surface area. The results suggest that an increased deformation occurs between the two layers of skin (as opposed to a single layer of skin and a hard probe surface), making it more difficult for the probe to reach the specified force. A second layer of skin also introduces a small increase in the amount of moisture present between the substrate, which will also reduce the bioadhesion parameters [11]. The disparity between the testing methods is also seen when comparing the homopolymeric HPMC gel to the binary gel, where adhesiveness and work of adhesion have significant differences when tested using a single-skin substrate (method one). These data strongly suggest that the addition of 6% PVP to the homopolymeric HPMC gel only favorably increases the bioadhesiveness, but does not affect the other mechanical properties. 

These differences between the two formulations are not apparent when tested using the two-skin layer method (method two). However, compared with method two, method one is closer to the clinical setting. 

There are currently no standardized methods for testing bioadhesiveness [10,11], and results vary significantly depending on different experimental setups. This finding emphasizes the need for a standardized method and testing setup for measuring bioadhesion parameters. Presently, it is hard to extrapolate results produced by different research groups under different settings.

Overall, this study contributes to the understanding of the properties and potential applications of binary hydrogels in topical products. The findings demonstrate the versatility and potential of binary hydrogels to improve adhesion, rheological properties, and drug delivery. Potential future directions in developing controlled-release gels should investigate using multiple polymers in gels and their ability to modulate the properties for their applications. This study highlights the ability of a binary gel to modify drug-release properties using parameters outside the classical models, notably, by affecting intermolecular binding. However, further research is needed to better understand and optimize the properties of binary hydrogels for specific applications.

## 5. Conclusions

The findings in this article demonstrate that the addition of PVP to HPMC to form binary gels provides an effective means of modifying the adhesive properties of the gel without significantly affecting other mechanical properties. This allows for a highly adhesive topical formulation that can still be applied on the skin comfortably while providing the controlled-release of drug over extended periods of time, which is highly desirable in many applications, such as the delivery of anti-inflammatories and chemotherapeutic agents for the treatment of skin conditions such as psoriasis and skin cancers. Furthermore, we demonstrate that binary hydrogels possess favorable pseudoplastic rheological properties that make them well-suited for topical applications. Our study also highlights the ability of binary hydrogels to enhance drug retention for sustained and controlled drug release. Finally, our findings reveal the importance of standardizing the methodology and setup for evaluating bioadhesion parameters. Overall, this research sheds light on the potential of PVP-HPMC binary hydrogels with low viscosity but high bioadhesiveness as a promising platform for sustained drug release in topical applications. When different drug molecules are incorporated, which may have different interactions with the polymers, further studies are needed to optimize the formulation. 

## Figures and Tables

**Figure 1 pharmaceutics-15-02360-f001:**
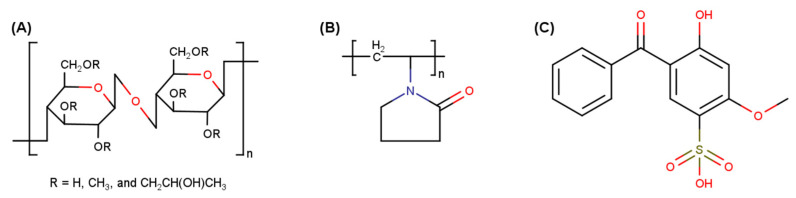
Chemical structures of (**A**) hydroxypropyl methylcellulose (HPMC), (**B**) polyvinylpyrrolidone (PVP), and (**C**) benzophenone-4.

**Figure 2 pharmaceutics-15-02360-f002:**
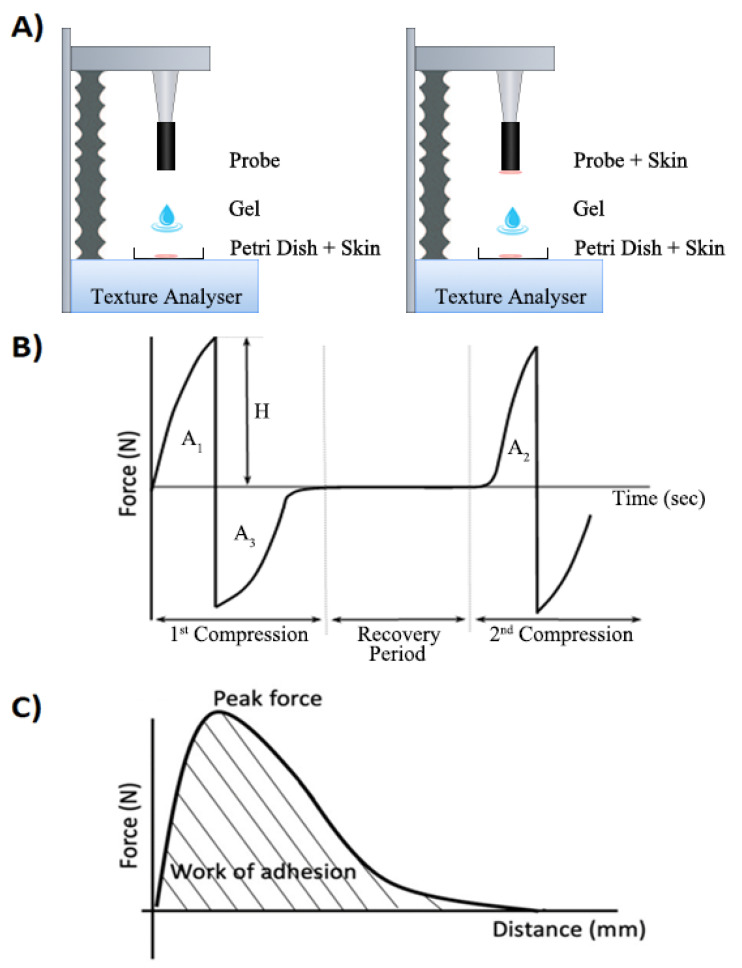
(**A**) Schematic diagram of the TXA setup for bioadhesion testing using a standard setup (**left**) with single skin substrate, or alternative setup (**right**) with gel between two skin substrates. (**B**) Force–time graphical output from texture profile analysis. H = Hardness. A_1_ = Compressibility. A_3_ = Adhesiveness. A_2_/A_1_ = Cohesiveness. (**C**) Schematic graph showing the applied peak force and work of adhesion provided by the texture analyzer software (Brookfield Rheocalc version 3.2.47).

**Figure 3 pharmaceutics-15-02360-f003:**
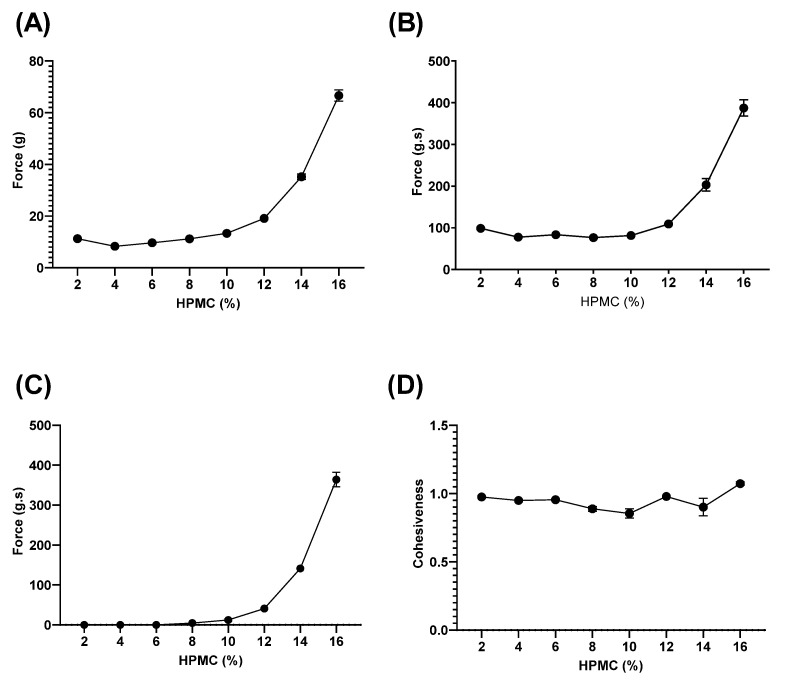
Mechanical properties of homopolymeric HPMC (2–16%) hydrogels obtained with a texture analyzer. (**A**) Hardness; (**B**) Compressibility; (**C**) Adhesiveness; and (**D**) Cohesiveness. Data are mean ± SD (n = 3).

**Figure 4 pharmaceutics-15-02360-f004:**
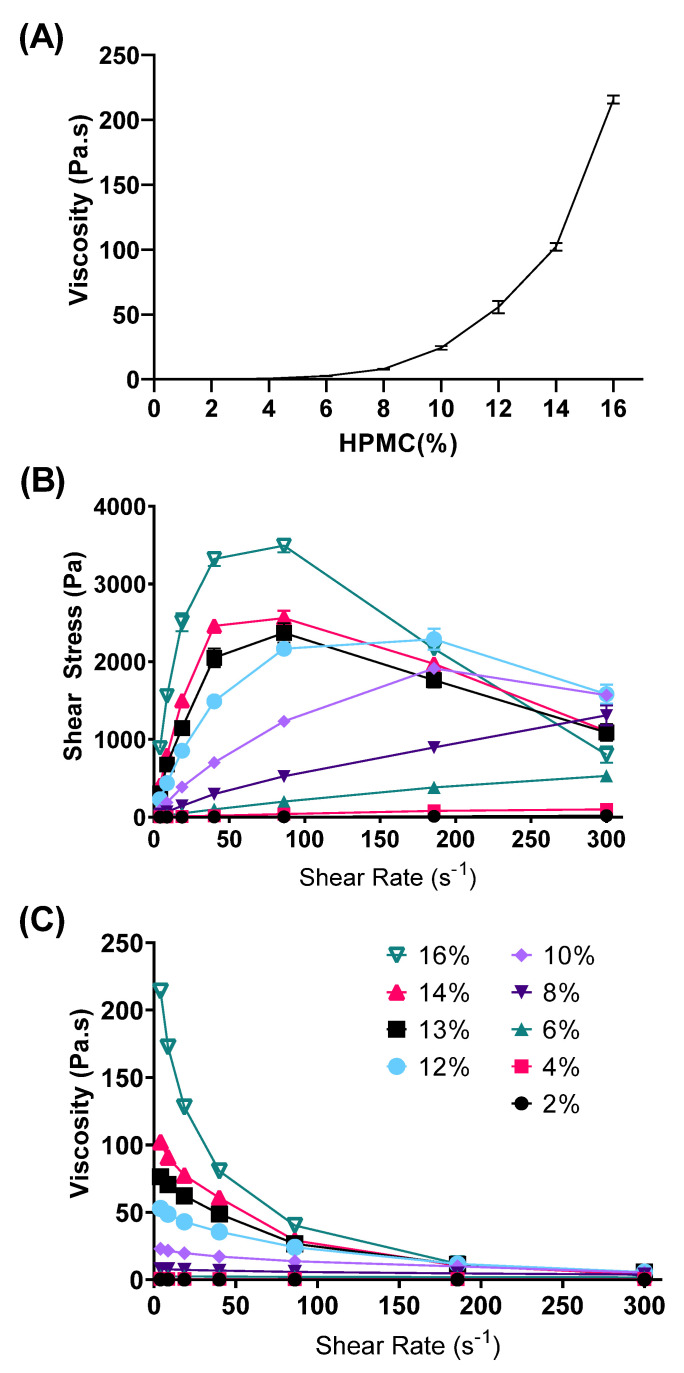
Rheological properties of gels containing HPMC 2–16%: (**A**) Viscosity curve at shear rate = 4 s^−1^, and rheograms of (**B**) shear stress vs. shear rate and (**C**) viscosity vs. shear rate. Data are mean ± SD (n = 3).

**Figure 5 pharmaceutics-15-02360-f005:**
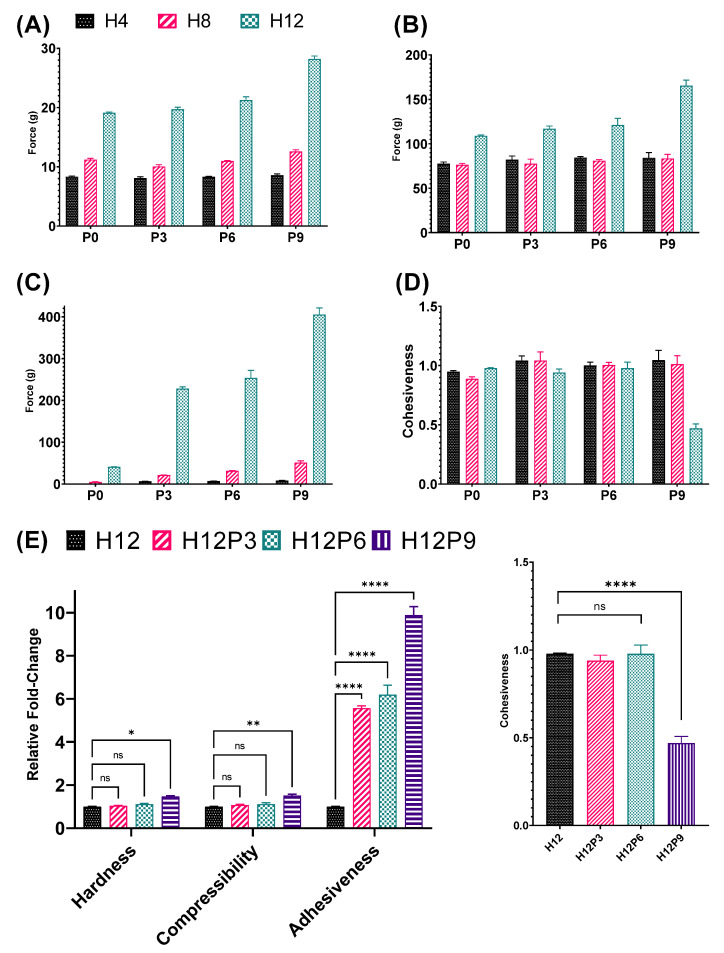
Physicomechanical properties of mono- or binary gels containing HPMC and PVP at various concentrations. (**A**) Hardness; (**B**) Compressibility; (**C**) Adhesiveness; and (**D**) Cohesiveness. Comparisons were made for the following: (**E**) hardness, compressibility, adhesiveness, and cohesiveness. The data show that the addition of PVP only increased the adhesiveness remarkably as a function of PVP concentration. Data are mean ± SD (n = 3). *: *p* < 0.05; **: *p* < 0.01; **** *p* < 0.0001; ns = non-significant difference.

**Figure 6 pharmaceutics-15-02360-f006:**
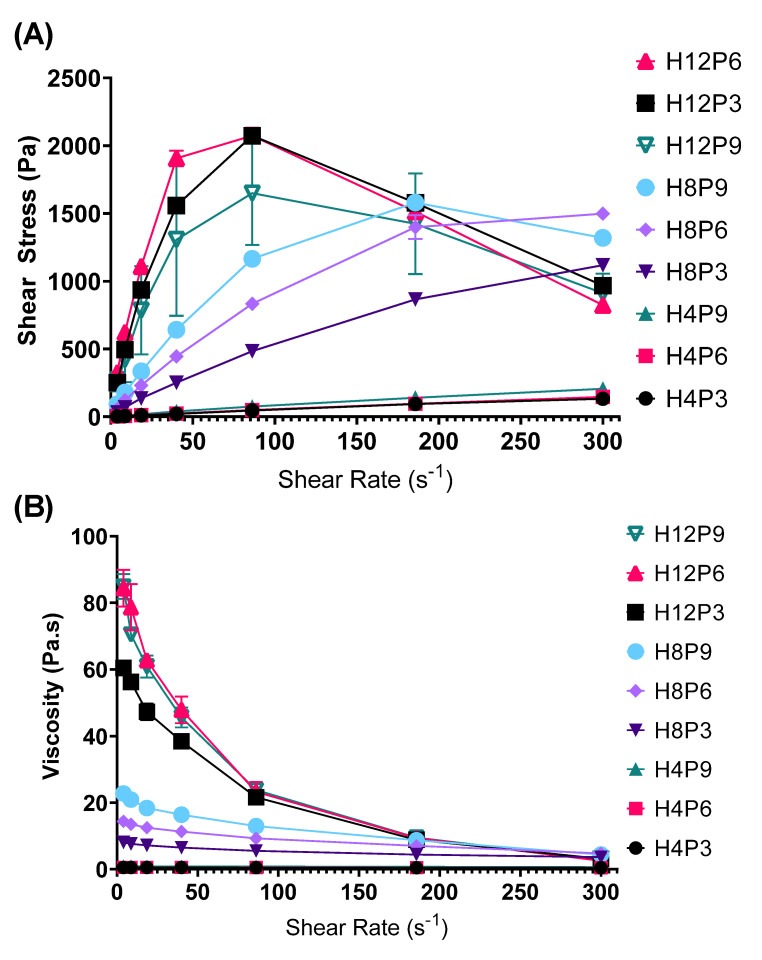
Plots of (**A**) shear stress vs. shear rate and (**B**) viscosity vs. shear rate for HPMC–PVP binary gel formulations. Data are means ± SD (n = 3).

**Figure 7 pharmaceutics-15-02360-f007:**
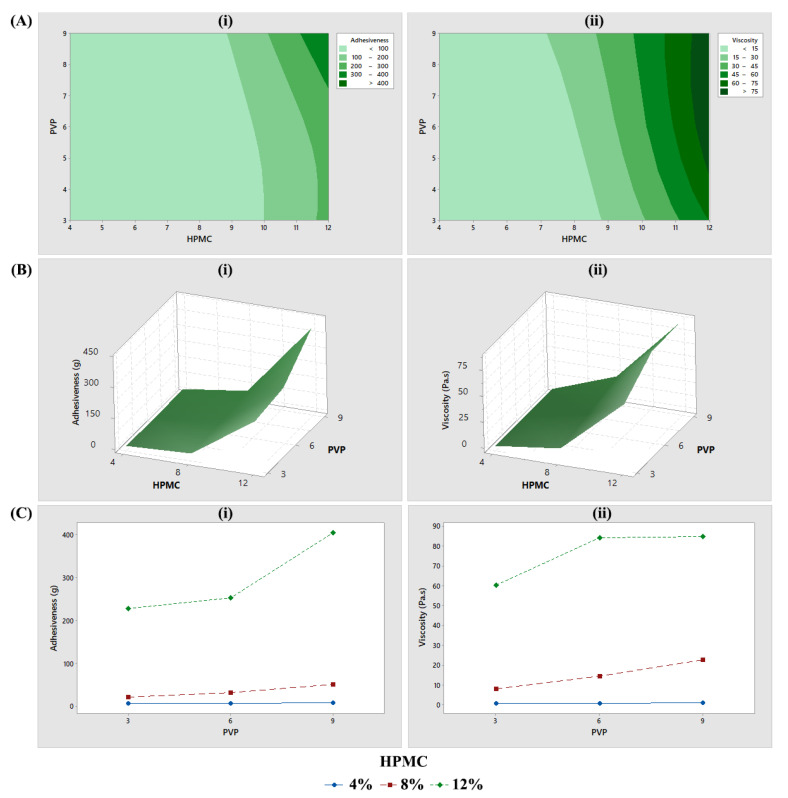
The 3^2^ factorial design analysis using Minitab. (**A**) Two-dimensional contour plot, (**B**) three-dimensional surface plot, (**C**) interaction plot, for adhesiveness (**i**) and viscosity (**ii**).

**Figure 8 pharmaceutics-15-02360-f008:**
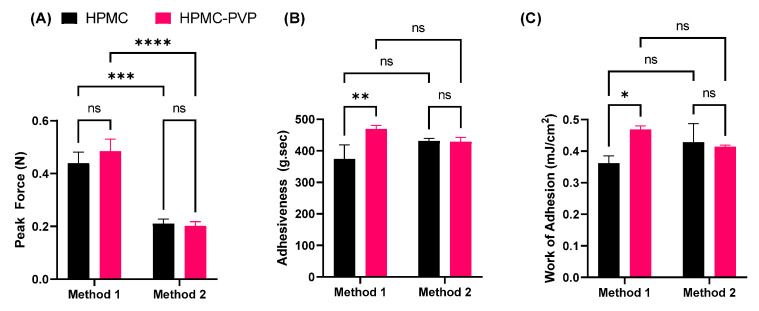
Bioadhesion results for the H12P6 binary gel for either a single (method one) or double (method two) skin substrate setups. (**A**) peak force; (**B**) adhesiveness; and (**C**) work of adhesion. Data are mean ± SD (n = 3). *: *p* < 0.05; **: *p* < 0.01; ***: *p* < 0.001; ****: *p* < 0.0001; ns = non-significant difference.

**Figure 9 pharmaceutics-15-02360-f009:**
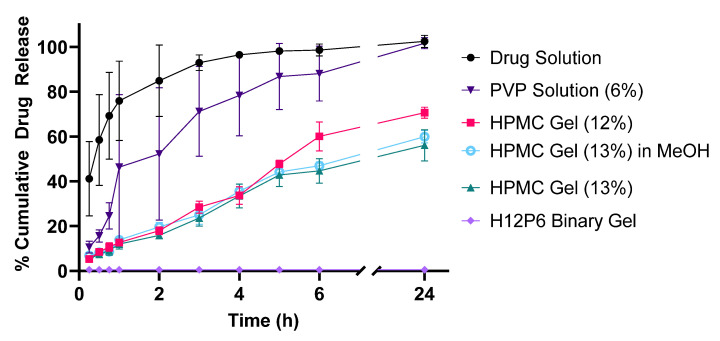
Cumulative drug release of benzophenone-4 over 24 h at 33 °C from formulation samples in dialysis bags. Drug release from the binary gel (H12P6) remained undetected throughout and is graphically represented as the limit of detection. All gel formulations contained 2.5% *w*/*w* of benzophenone-4. Data are mean ± SD (n = 3).

**Figure 10 pharmaceutics-15-02360-f010:**
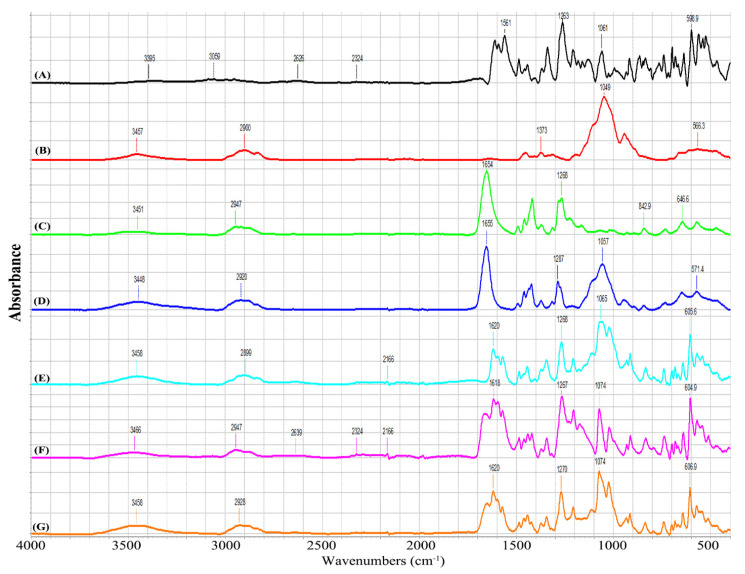
FTIR Spectra for (**A**) benzophenone-4, (**B**) HPMC, (**C**) PVP, (**D**) HPMC–PVP, (**E**) HPMC + BNZ4, (**F**) PVP + BNZ4, and (**G**) HPMC–PVP + BNZ4 freeze-dried gels.

**Figure 11 pharmaceutics-15-02360-f011:**
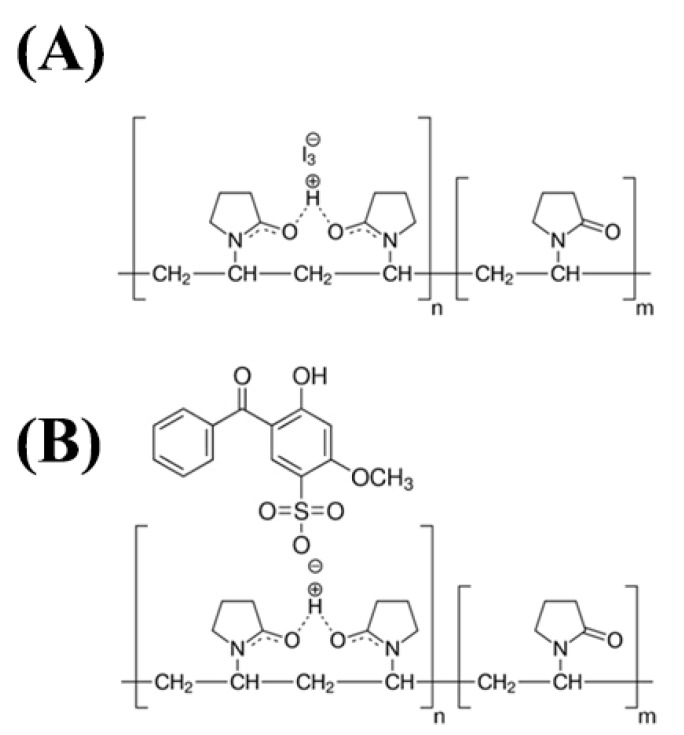
(**A**) Electrostatic adsorption of triiodide by PVP. (**B**) Possible electrostatic adsorption of benzophenone-4 by PVP which retarded the drug release from the gel.

**Table 1 pharmaceutics-15-02360-t001:** Amounts of HPMC (X_1_) and PVP (X_2_) powders in the binary gels generated by a 3^2^ factorial design.

Gel Number	1	2	3	4	5	6	7	8	9
X_1_: HMPC (% *w*/*v*)	4	4	4	8	8	8	12	12	12
X_2_: PVP (% *w*/*v*)	3	6	9	3	6	9	3	6	9

## Data Availability

Data is contained within the article. The data presented in this publication are available upon reasonable request from the corresponding authors.

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
