# Peer review of "Hydroxypropyl Methylcellulose Bioadhesive Hydrogels for Topical Application and Sustained Drug Release: The Effect of Polyvinylpyrrolidone on the Physicomechanical Properties of Hydrogel"

_pharmaceutics, 2023, doi:10.3390/pharmaceutics15092360_

Round 1
Reviewer 1 Report
1. The introduction should provide more context and background information on the current challenges in drug delivery for skin conditions such as psoriasis and skin cancers. This will help readers understand the significance of the research and its potential impact.
2. Explain the rationale behind choosing a three-level factorial design for forming the binary hydrogels.
3. Discuss the observed intermolecular interactions within the polymers and their implications on the gel properties.
4. Discuss potential future directions or applications for this research.
5. The introduction should provide more context and background information on the current challenges in drug delivery for skin conditions such as psoriasis and skin cancers. This will help readers understand the significance of the research and its potential impact.
Reviewer 2 Report
The addition of polyvinylpyrrolidone (PVC) to hydroxypropyl methylcellulose (HPMC) allowed the obtaining of binary gels, with improved adhesiveness, without significantly affecting other properties (hardness, shear-thinning feature and viscosity). These gels were used for sustained drug release over 24 h at 33 °C. Thus, the release of the drug (benzo-phenone-4) in these binary hydrogels followed a zero-order kinetics, with drug release significantly retarded by the presence of PVP, likely due to intermolecular interactions between the drug and the binary polymer as confirmed by FTIR.
The manuscript is relevant and the topic is original. The conclusions are consistent with the evidence and arguments presented. The literature is up-dated.
The manuscript is well written and the text is clear and easy to read.
I agree with its publication in actual form.
Reviewer 3 Report
In ‘Hydroxypropyl methylcellulose Bioadhesive Hydrogels for Topical Application and Sustained Drug Release: The Effect of polyvinylpyrrolidone on Hydrogel Physicomechanical Properties’, Pan et al. prepared HPMC/PVP hydrogels in a range of concentrations, using a factorial design approach, and measured their physicomechanical and rheological properties as well as their adhesiveness to porcine skin and drug release profiles. Overall, the texture analysis and measurement of rheological properties is rather thorough, but I have the following comments.
Major comments:
- The ex vivo bioadhesion experiments are interesting but they lack a control or reference measurement (like a material known to have good adhesion to skin) to be able to understand if the results are meaningful.
- The drug release experiments are minimal. I can understand why one would choose one formulation to test in the ex vivo experiments, but more formulations could be tested in the drug release experiments. It is difficult to understand how the release is so fast in the mono-component gels, but then it drops to no release for the binary gels. It would be interesting to see if the other formulations have an intermediate release rate.
Minor comments:
- typo in Figure 4b caption (shear stress instead of shear rate)
- p. 9: it seems like the adhesiveness increases significantly already at 3% and 6%, not 9%
- Figure 5 a-d: error bars are missing
- The release curves for the HPMC gels are mostly overlapping, so it is difficult to understand how there is a difference between 12% and 13% but not a difference between PBS and methanol.
- There are some typos and incomplete sentences.
Round 2
Reviewer 1 Report
Everything is ok